# The Emerging Roles of Intracellular PCSK9 and Their Implications in Endoplasmic Reticulum Stress and Metabolic Diseases

**DOI:** 10.3390/metabo12030215

**Published:** 2022-02-26

**Authors:** Paul F. Lebeau, Khrystyna Platko, Jae Hyun Byun, Yumna Makda, Richard C. Austin

**Affiliations:** Department of Medicine, Division of Nephrology, Institute of St. Joe’s Hamilton, The Hamilton Center for Kidney Research, McMaster University, Hamilton, ON L8N 4A6, Canada; paulebeau91@gmail.com (P.F.L.); platkok@gmail.com (K.P.); byunjh@mcmaster.ca (J.H.B.); makdayr@mcmaster.ca (Y.M.)

**Keywords:** endoplasmic reticulum, loss-of-function, chaperone, CD36, liver disease, kidney disease, neurodegenerative disease, inflammatory disease

## Abstract

The importance of the proprotein convertase subtilisin/kexin type-9 (*PCSK9*) gene was quickly recognized by the scientific community as the third locus for familial hypercholesterolemia. By promoting the degradation of the low-density lipoprotein receptor (LDLR), secreted PCSK9 protein plays a vital role in the regulation of circulating cholesterol levels and cardiovascular disease risk. For this reason, the majority of published works have focused on the secreted form of PCSK9 since its initial characterization in 2003. In recent years, however, PCSK9 has been shown to play roles in a variety of cellular pathways and disease contexts in LDLR-dependent and -independent manners. This article examines the current body of literature that uncovers the intracellular and LDLR-independent roles of PCSK9 and also explores the many downstream implications in metabolic diseases.

## 1. Introduction

Widely considered as the greatest advancement in our understanding of cardiovascular disease (CVD) since the discovery of the low-density lipoprotein (LDL) receptor (*LDLR*) over 40 years ago, the characterization of the proprotein convertase subtilisin/kexin type-9 (*PCSK9*) gene and its encoded protein product was published in two back-to-back articles by Seidah et al. and Abifadel et al. in 2003 [1,2]. In the years that followed, it was also reported that loss-of-function (LOF) variants in *PCSK9* were associated with a lifelong state of hypocholesterolemia and a substantial reduction in CVD risk [3,4,5,6,7]. Nearly 20 years later, our understanding of PCSK9 has evolved, but its primary role as a regulator of circulating cholesterol levels has not changed. The major function of PCSK9 is to enhance the degradation of the LDLR and reduce the clearance of LDL cholesterol (LDLc) from the circulation, where it otherwise increases CVD risk [8,9,10]. Since the time of these seminal findings, PCSK9 has been shown to regulate the expression of a variety of intracellular and cell-surface proteins and potentially play a role in a variety of diseases [11]. As a secretory protein and third locus of familial hypercholesterolemia, the majority of reports have focused on the role of circulating PCSK9 in the context of CVD.

In this review, we will focus on the recent studies that have characterized the intracellular roles of PCSK9. As a secretory protein, the PCSK9 lifecycle begins in the endoplasmic reticulum (ER), where it undergoes several post-translational modifications and interacts with a variety of ER-resident proteins. Although wild-type PCSK9 readily transits the ER, many ER-retained LOF variants have now been identified and new therapeutic modalities that cause ER retention are also being explored. In addition, this review will focus on new advances on the effects of PCSK9, both intracellular and extracellular, on metabolic diseases such as liver disease, chronic kidney disease (CKD), and neurodegenerative disorders.

## 2. PCSK9 and Its Loss-of-Function Variants in the Endoplasmic Reticulum

PCSK9 is primarily expressed in the liver, small intestine, and kidney where it is synthesized as a 692-amino-acid zymogen [2]. The zymogen consists of a signal peptide (aa 1–30), which promotes its entry into the ER via the translocon (Figure 1). Following entry, the prosegment (aa 31–152) is autocatalytically cleaved by the catalytic domain (aa 153–404) at position Q152↓ [1,12]. Similar to other proprotein convertases [13], the newly-cleaved PCSK9 prosegment forms a covalent bond with the catalytic domain and blocks any further catalytic activity of the protein and also acts as an intramolecular chaperone that promotes proper folding and ER exit [1]. The PCSK9 zymogen also consists of a hinge region (aa 405–451) and a C-terminal cysteine- and histidine-rich domain (aa 452–692), which play important roles in LDLR binding and cellular trafficking [14]. The C-terminal domain of PCSK9, in particular, consists of three tandem repeats with structural similarities, which are referred to as the M1, M2, and M3 modules. Due to the density of His residues in the M2 domain, it is thought that the C-terminal domain plays a significant pH-dependent role in the sorting of PCSK9 to the endosome [15,16].

The C-terminal domain of PCSK9 is unique among the proprotein convertase family due to its net positive charge with several protein–protein interaction motifs [15]. Deletion of the C-terminal domain of PCSK9 also impairs PCSK9 secretion, likely via failure of the truncated protein to interact with coat protein complex II (COPII) component (SEC24) [14,17]. Following its autocatalytic cleavage in the ER, PCSK9 is shuttled to the Golgi via the canonical COPII vesicle pathway. The knockdown of SEC24 isoforms A, B, and C—key components of the COPII vesicle—has been shown to significantly reduce PCSK9 secretion in cultured hepatocytes. In the same study, deletion of amino acids 445 to 692 also impaired PCSK9 maturation and its exit from the ER. To reach SEC24 on the cytoplasmic side of the ER, however, ER-luminal PCSK9 is first sorted by the ER cargo receptor, Surf4 [18]. This model was characterized using a proximity-dependent biotinylation approach in combination with co-immunoprecipitation studies. Emmer and colleagues demonstrated that PCSK9 interacted directly with the Surf4 ER cargo receptor and that knockdown of Surf4 significantly reduced overexpressed PCSK9 secretion in Hek293 cells [18]. Recent studies, however, in HepG2 cells, HuH7 cells, as well as in a mouse model of hepatic Surf4 knockdown, have failed to corroborate these findings [19,20]. It is likely, however, that a diversity of ER cargo receptors contributes to the exit of PCSK9 from the ER, and that such mechanisms vary between cell types. Clearly, additional studies are required in order to better understand how PCSK9 exits the ER and is sorted into COPII vesicles. Interestingly, dynamin-related protein-1 (DRP1), which contributes to ER remodeling, was recently shown to play a role in the secretion of PCSK9; treatment of HepG2 cells with the mitochondrial division inhibitor-1, as well as hepatic knockout of DRP1, significantly reduced secreted PCSK9 levels in cells and mice [21].

The expression and secretion of PCSK9 from secretory cells is dependent on the ER for more than the process of autocatalytic cleavage. PCSK9 was first shown to be phosphorylated on serine residues at positions 47 and 688 by a Golgi casein kinase [22], and at serines 47, 666, 668, and 688 by Fam20C in the ER [23]. These gain-of-function (GOF) phosphorylation events have been shown to promote the secretion of PCSK9 from hepatocytes, reduce PCSK9 proteolysis, and enhance the degradation of the LDLR. PCSK9 is also N-glycosylated at position 533 and sulfonated on tyrosine residues [12,24]. Consistent with the phosphorylation events, the glycosylation and sulfonation of PCSK9 were shown not to be necessary for PCSK9 function but were shown to enhance PCSK9-induced LDLR degradation.

The sterol regulatory element-binding protein-2 (SREBP2), which is the primary transcriptional driver of the *de novo* synthesis of PCSK9 [25,26,27], also resides in the ER in a pre-mature form [28,29]. During conditions of low-intracellular cholesterol, SREBP2 is activated and shuttled to the nucleus where it induces the expression of many cholesterol regulatory genes, including *PCSK9*, *LDLR*, and *3-hydroxy-3-methylglutyryl-coenzyme A reductase* (*HMGR*) [29]. ER expansion and the need for new cholesterol-rich ER membranes is known to mitigate ER stress and may be the reason that SREBP2 is activated by such conditions [30,31,32]. Recent evidence also demonstrates that ER Ca^2+^ levels, which play a significant role in ER chaperone activity, modulate SREBP2 activation via GRP78 to control PCSK9 expression and circulating LDLc levels [33]. In this study, exposure to caffeine was found to increase hepatic ER Ca^2+^ levels, block *de novo* synthesis, and reduce circulating levels of PCSK9 in vitro, in vivo, and in healthy volunteers [33]. Additional transcription factors and promoters of *de novo* PCSK9 synthesis, such as the hepatocyte nuclear factor (HNF1α) and SREBP1, are also known to play a role in ER stress [31,34,35,36]. Although small molecule antagonists of PCSK9 autocatalytic cleavage that cause ER retention are highly desirable as a treatment strategy for CVD, this approach is not without its challenges. Unlike other druggable proteases, PCSK9 is unique in that it is only known to undergo a single proteolytic action upon itself [37]. Following this event, which is thought to take place with zero-order kinetics following zymogen entry into the ER, the PCSK9 maturation process is largely complete, and the catalytic site is inaccessible to other substrates [38]. Furthermore, the process of PCSK9 autocatalytic cleavage in the ER is protected from potential inhibitors by two lipid bilayers. Overall, PCSK9 plays a significant role on the regulation of circulating LDLc and CVD risk, and the inhibition of PCSK9 autocatalytic cleavage remains an important treatment avenue [1].

The clinical benefit of LOF variants in PCSK9 was first described following the sequencing of *PCSK9* in African American individuals with hypocholesterolemia from the Dallas Heart Study. Among the African American patients (n = 3363) involved in the study, 2.6% carried a heterozygous non-sense PCSK9 variant (Y142X or C679X), which led to a 28% reduction in circulating LDLc and was associated with an 88% reduction in coronary heart disease (CHD) [4]. Similarly, the R46L variant was identified in 3.2% of the white population (n = 9524) included in the study, which led to a 15% reduction in LDLc and a 47% reduction in CHD [4]. To date, >20 LOF PCSK9 variants have been reported occuringin all domains of the protein [39]. In contrast, GOF variants in *PCSK9* were first correlated with autosomal dominant hypercholesterolemia in 2003 following the seminal discovery and characterization of the gene and encoded protein [1,2]. While the majority of GOF variants impact later stages of the PCSK9 lifecycle, such as increased LDLR binding, most of the LOF mutations reduce the exit of PCSK9 from the cell during early stages of protein maturation. A reduction or abolishment of PCSK9 autocatalytic cleavage, for instance, results in a LOF PCSK9 protein. Mutations in the site of cleavage, such as Q152H, as well as mutations in the catalytic domain, such as L253F, N354I, and H391N, prevent the autocatalytic cleavage and maturation of PCSK9 [40,41,42]. Because residues of the prodomain regulate proteolysis and secretion independently, it is also likely that such LOF variants impact the exit of PCSK9 from the ER in an additive or even synergistic manner [38]. Likely due to the ability of PCSK9 to oligomerize in the ER [1], the overexpression of ER-retained variants of PCSK9 also blocks the exit of endogenous wild-type PCSK9 [5]. This phenomenon was also observed in mice, as well as in subjects expressing the PCSK9-Q152H variant [43]. Mutations in the signal peptide, such as L10 and 61_63dupCTG, have been shown to impact the trafficking of PCSK9, likely resulting in ER retention as well [44,45]. The relatively common R46L prodomain variant, however, is not thought to lead to ER retention. A reduction in the GOF phosphorylation of PCSK9 in individuals with R46L results in increased proteolytic degradation of the variant, which has 50% reduced affinity for the LDLR [22]. Additionally, mutations in the prodomain (R97del, R104C, G106R, and V114A), as well as the C-terminal region (R434W, S462P, and C679X) have also been shown to result in ER retention of PCSK9 [40,42,46,47,48,49].

The good health and longevity reported in individuals with LOF *PCSK9* variants strongly suggests that such mutations are not deleterious [5,43]. In fact, hypocholesterolemia accompanied by a marked reduction in CHD risk, are among the few outcomes reported in these individuals to date [5,43]. It is intriguing, however, that the retention of PCSK9 in the ER of secretory or endocrine cells, such as hepatocytes, does not lead to ER storage disease (ERSD) [43]. Many heritable LOF variants in cell-surface or secretory proteins that transit the ER and are retained due to failure to pass ER quality controls lead to ERSD. Mutations in alpha (1)-antitrypsin Z, for instance, induce chronic ER stress and liver cirrhosis, as well as hepatocellular carcinoma [50,51]. Additional examples include arginine-vasopressin mutations (G14R and G17V) involved in familial neurohypophyseal diabetes insipidus [52,53], cystic fibrosis transmembrane conductance regulator-ΔF508 involved in cystic fibrosis [54], thyroglobulin mutations involved in congenital goiter and hyperthyroidism [55], and mutations in the Notch receptor known to promote cerebral autosomal-dominant arteriopathy and leukoencephalopathy [56].

## 3. PCSK9 as a Putative Co-Chaperone of the ER

Hepatocytes, like all secretory cells, are rich in ER and are sensitive to disturbances that affect ER homeostasis. Because the majority of cell-surface and secretory proteins transit this organelle, it is vitally important for hepatocytes to maintain ER function [57,58]. The ER consists of multiple interdigitated cisternae located in the perinuclear region of the cell and can be subdivided into the ribosome-enriched rough ER responsible for *de novo* protein synthesis and maturation, as well as the smooth ER responsible for the synthesis of lipids and steroids [31]. In order to carry out such a diversity of tasks, the ER contains an abundance of resident molecular chaperones within its lumen. Chaperones are charged with the task of ensuring that the constant influx of nascent proteins entering the ER are properly folded before exit and subsequent secretion from the cell. When the influx of newly synthesized proteins entering the ER exceeds its protein folding capacity, an accumulation of misfolded polypeptides ensues. This state is referred to as ER stress, and often occurs when a chemical imbalance within the luminal environment reduces ER-resident chaperone protein folding efficiency. Given the importance of the ER in all secretory and metabolically active tissues, it is not surprising that ER stress has now been identified as a driver in the development of a range of human diseases—including CVD, liver disease, CKD, as well as neurodegenerative diseases, to name a few [57,59,60,61].

While in the ER, PCSK9 is known to interact with the ER-resident chaperone glucose-regulated protein of 94 kDa (GRP94) [43,62,63] (Figure 2). In contrast to many other secretory proteins that transit the ER, however, the processing and maturation of PCSK9 is not dependent on its interaction with this chaperone. Rather, it is hypothesized that ER-resident GRP94 acts as an antagonist of the PCSK9-LDLR complex, thereby preventing the intracellular route of LDLR degradation, promoting cell-surface LDLR expression, and reducing circulating LDLc levels [62]. Due to its dependence on SREBP2, an ER stress-inducible transcription factor, PCSK9 expression was shown to be induced by ER stress caused by ER Ca^2+^ depletion [63]. ER stress, however, also led to the retention of PCSK9 in the ER [63]. This behavior parallels that of the many other chaperones that are highly expressed and abundant in the ER during conditions of ER stress.

ER PCSK9 accumulation, in contrast to the secretory proteins known to cause ERSD, does not cause ER stress in the livers of mice and in cultured hepatocytes due to its ability to interact with GRP94 [43,64]. In contrast, the expression of an ER-retained LOF LDLR-G544V variant caused robust hepatic ER stress, apoptosis, and liver injury [43]. Interestingly, the expression of the ER-retained LOF PCSK9-Q152H variant led to ER stress and cytotoxicity only in GRP94 knockdown cells [64]. Results of this study suggest that GRP94 shields ER-resident PCSK9 from the glucose-regulated protein of 78 kDa (GRP78), which acts as the sensor of the unfolded protein response (UPR) [64]. Recently published clinical evidence also demonstrated that subjects expressing the PCSK9-Q152H variant are healthy, despite a lifelong state of hypocholesterolemia and hepatic ER-PCSK9 accumulation [43]. In vitro and in vivo experiments from this study demonstrated that ER retention of PCSK9 increased the protein abundance of ER-resident chaperones GRP78 and GRP94, thereby bestowing increased resistance against hepatic ER stress. Importantly, this effect is not unique to PCSK9. The BAG family molecular chaperone regulator 5 (Bag5) and the polypeptide N-acetylgalactosaminyltransferase 6 (GALNT6) have also been shown to protect against ER stress by increasing the protein abundance of ER-resident chaperones [66,67]. Consistent with the notion that PCSK9 may act as a putative co-chaperone of the ER, a previous study also demonstrated that ER-resident PCSK9 acts as a chaperone of newly-synthesized LDLR [65]. In this study, wild-type PCSK9 was shown to promote the transit of the LDLR to the cell surface, while cleavage variants that did not exit the ER failed to do so [65]. Overall, accumulating evidence demonstrates that ER-resident PCSK9 can modulate chaperones such as GRP94, as well as cell-surface receptors such as the LDLR. Given the roles of such proteins in the context of CVD and other metabolic disorders, the long-term effects of therapies that block *de novo* synthesis of PCSK9 protein remain to be elucidated.

## 4. PCSK9 and Liver Disease

Non-alcoholic fatty liver disease (NAFLD) is one of the most prevalent hepatic disorders worldwide, affecting 25–30% of the total population [68]. Clinically, NAFLD is characterized by an accumulation of triglycerides in the liver that exceeds 5% of total liver weight and is comprised of non-alcoholic fatty liver (NAFL) and non-alcoholic steatohepatitis (NASH). While the hallmark of NAFL is fat accumulation in the liver, NASH exhibits the addition of inflammation and hepatocyte injury, and can progress to cirrhosis and eventually hepatocellular carcinoma and end stage liver failure [69]. Although the exact etiology of this disease is poorly understood, it is known that the pathophysiology of NAFLD involves insulin resistance which promotes hepatic steatosis, a process enhanced by obesity and type 2 diabetes (T2D) [70]. An increasing body of evidence shows that impaired liver function is an independent risk factor for several metabolic conditions, including CVD [71], insulin resistance [70], and CKD [72,73]. Warranted by the complexity of this disease, numerous underlying molecular drivers of NAFLD have been identified and are described in detail by Friedman and colleagues [74]. However, recent findings point to the importance of PCSK9, traditionally regarded as a modulator of CVD, in the regulation of hepatic lipid homeostasis.

Indeed, pre-clinical and clinical data have highlighted a potential link between PCSK9 and NAFLD, demonstrating that circulating PCSK9 can prevent hepatic lipid uptake and accumulation. High fat diet (HFD) feeding was shown to induce hepatic steatosis and increase circulating as well as hepatic PCSK9 levels in mice [75]. In their study, Demers et al. show that PCSK9 regulates the expression of a well-established modulator of fatty acid (FA) uptake and driver of hepatic steatosis, cluster of differentiation (CD36) [76]. Consistent with these observations, Lebeau et al. demonstrated that in immortalized hepatocytes, PCSK9 regulates FA uptake in a CD36-dependent manner [77]. Furthermore, *Pcsk9^−/−^* mice exhibit increased hepatic lipid accumulation and CD36 expression [76,77]. In line with these findings, HFD caused severe hepatic steatosis, ER stress, inflammation, and fibrosis in the livers of *Pcsk9^−/−^* mice [77]. Demers and colleagues were the first to show that PCSK9 could promote the degradation of CD36 by interacting with the extracellular loop and mediating its internalization [76]. In contrast, the domain of PCSK9 that interacts with CD36 remains to be determined. PCSK9-mediated CD36 degradation is thought to occur via the extracellular and intracellular pathways, where PCSK9 impedes the recycling of the receptor to the cell surface and induces its degradation within the lysosome [76]. This process was shown to reduce the uptake of long-chain fatty acids, oxidized LDL (oxLDL), and triglycerides. Using in vitro and in vivo models of PCSK9 inhibition, Demers et al. reported increased hepatic triglyceride content, suggesting that elevated CD36 levels in the liver may lead to increased susceptibility to NAFLD [76]. The discovery that *PCSK9* gene deletion promotes murine NASH was also corroborated in a recent study by Ioannou and colleagues. Upon feeding of a high fat/high cholesterol diet for 9 months, *Pcsk9**^−/−^* mice exhibited increased levels of hepatic cholesterol crystallization, increased crown-like structures of macrophages, higher levels of apoptosis and inflammation, as well as 11-fold increase in hepatic fibrosis compared to controls [78]. Interestingly, monoclonal antibodies against PCSK9 that are currently available to patients as an adjunct to diet, alone, or in combination with other lipid-lowering therapies, increase circulating levels of the protein as a direct result of preventing its interaction/degradation with the LDLR [79]. Although this antibody-bound form of PCSK9 is no longer capable of interacting with the LDLR, its ability to interact with, and modulate, the expression of CD36 remains to be elucidated.

Interestingly, adenovirus-mediated overexpression of the ER-retained PCSK9-Q152H LOF variant unexpectedly protected against ER stress and liver injury rather than inducing them in mice [43]. Conflicting evidence, however, suggests that liver-specific overexpression of human PCSK9 in mice drives NAFLD and fibrosis upon a dietary challenge [80]. This suggests that PCSK9 may play a dual role depending on its cellular localization. Additionally, E2f1 has been identified as one of the major regulators of PCSK9 expression, and *E2f1^−/−^* mice fed a high-cholesterol diet (HCD) displayed increased hepatic lipid accumulation and fibrosis. Importantly, the NAFLD phenotype was reversed by re-expression of PCSK9 in the liver [81]. Collectively, these studies reinforce the importance of PCSK9 in the regulation of hepatic lipid metabolism/homeostasis.

Although most pre-clinical evidence supports a pro-steatotic role of PCSK9, clinical observations remain controversial. In morbidly obese patients, hepatic PCSK9 expression is inversely correlated with hepatic fat accumulation, while circulating PCSK9 levels correlate with the severity of liver steatosis [82]. In contrast, Ruscica and colleagues demonstrated that hepatic PCSK9 expression and circulating PCSK9 levels correlate with steatosis in morbidly obese patients who have undergone bariatric surgery [83]. In support of this notion, a recent study found that hepatic expression of PCSK9 increases with severity of steatosis [80]. In a separate study, no association between circulating PCSK9 and liver enzymes was shown in obese patients [84]. Also in conflict with previous observations, no association between liver fat and circulating PCSK9 or hepatic PCSK9 mRNA expression was reported in obese patients [84]. To delineate the specific role of PCSK9 on hepatic health, three separate studies examined the effect of LOF PCSK9 variants on hepatic steatosis and liver function [43,80,85]. Similar to findings in mice, carriers of the PCSK9-Q152H variant exhibited normal liver function despite their lifelong state of PCSK9 retention [43]; perhaps suggesting that the PCSK9-Q152H variant attenuates liver damage. On the other hand, early investigations on the effect of PCSK9-R46L showed that carriers of this variant displayed a two-fold increase in the prevalence of hepatic steatosis [85], while more recent data suggest that the PCSK9-R46L variant is protective against liver damage in patients with NAFLD [80]. Thus, the direction of the relationship between *PCSK9* and NAFLD in the context of human disease remains unclear. Collectively, there are a variety of causes of liver disease, and it is likely that PCSK9 expression correlates with certain forms of the disease (i.e., diet-induced liver disease). It is also likely that PCSK9 expression correlates with liver disease only during certain stages of the disease. It is not surprising, therefore, that results from studies based on a diversity of patient populations are not well aligned.

## 5. PCSK9 and Cardio-Renal Syndrome

CKD is a prevalent, chronic disease that is primarily characterized by a decline in renal function, and often presents with CVD [86]. As such, an increasing number of studies have investigated the association between PCSK9 and renal function. Konarzewski et al. first reported a two-fold increase in circulating PCSK9 levels in CKD patients (n = 44), with a significant negative correlation between PCSK9 and estimated glomerular filtration rate (eGFR) [87]. Other groups, however, did not report significant correlations between PCSK9 and eGFR in patients from two independent long-term trials, which aimed to characterize cardio–renal interactions based on statin intake [88]. The absence of a significant correlation was also observed in nondiabetic patients diagnosed during a variety of stages of CKD. Similarly, Morena et al. reported a correlation between PCSK9 and apolipoproteinB (ApoB), but failed to observe a correlation between PCSK9 and eGFR or proteinuria [89]. Interestingly, however, vaccination against PCSK9 protected against renal disease in mice by modulating fatty acid oxidation in the kidney. Compared to the Qβ viral control, the PCSK9Qβ vaccine protected against renal lipid accumulation and fibrosis in both unilateral ureteral obstruction (UUO) and N-nitro-L-arginine methyl ester models [90].

Recent research has also been conducted on the intracellular roles of PCSK9 in the kidney. Using an adriamycin nephropathy model, Zhang et al. reported that renal inflammation and subsequent lipid deposition was associated with a downregulation in renal PCSK9 expression [91]. In turn, this impacted the expression of receptors known to promote lipid uptake from the circulation. It was observed that renal HNF1α levels were suppressed, while SREBP2 and the LDLR were upregulated, suggesting an important role of HNF1α in the regulation of renal PCSK9. Barisione et al. have also reported that renal PCSK9 expression is impacted by ischemia [92]. Using a rat model, ischemia was shown to impact both renal and circulating levels of PCSK9, strengthening the notion that PCSK9 may also be an important target for the management of cardio-renal syndrome [92].

Others have investigated the association between PCSK9 and blood pressure. Sharotri et al. first reported a negative association between PCSK9 and the epithelial Na(+) channel (EnaC), which plays a critical role in blood pressure regulation by maintaining sodium homeostasis [93]. Contrary to these in vitro findings, however, Berger et al. reported no significant association between PCSK9 deficiency and changes in blood pressure that were tested in two well-established hypertensive mouse models; including the administration of (1) NG-Nitro-L-arginine-methyl ester (l-NAME), and (2) angiotensin II with deoxycorticosterone acetate-salt (DOCA) [94]. Interestingly, although an active subunit of EnaC was significantly higher in *Pcsk9^−/−^* mice, these mice continued to exhibit similar urinary sodium excretion and blood pressure levels compared to wild-type mice.

## 6. PCSK9 and Neurodegenerative Disease

First known for its role in neuronal apoptosis, PCSK9 is highly expressed in the telencephalon neurons, promoting neuronal differentiation and regulating cellular apoptosis during neurogenesis [1]. Additionally, dysregulated lipid homeostasis is strongly associated with the progression of several neurodegenerative diseases due to the cholesterol-dependence of the brain [95]. Compromised lipid metabolism has been shown to promote the denervation of neuromuscular junctions, impair neuronal transport, and promote cytoskeletal and mitochondrial dysfunction [96]. As such, several studies have been conducted to explore the potential role that PCSK9 may have in the progression of neurodegenerative diseases.

Wu et al. first reported that PCSK9 promotes neuronal apoptosis by upregulating caspases and reducing apolipoprotein E receptor 2 (ApoER2) expression [97]. Consistent with this observation, PCSK9 silencing in mice protected against cerebral ischemia-induced neuronal apoptosis, thereby preventing the progression of brain injury [98]. Interestingly, the association between PCSK9 and Alzheimer’s disease (AD) in the literature remains unclear. One of the canonical features of AD pathology is the accumulation of amyloid β (Aβ) plaque as a result of changes in the pathways responsible for Aβ; namely, the pathways regulated by the amyloid precursor protein (APP) and β secretase-1 (BACE1) [99]. BACE1, in particular, is known to play a major role in the development of AD due to its ability to clear Aβ. Studies investigating AD pathophysiology in the last decade have uncovered a significant connection between AD and dyslipidemia [99,100]. Interestingly, PCSK9 was shown to decrease cholesterol uptake in the brain by degrading the LDLR-related protein-1 (LRP1), and subsequently decreasing Aβ clearance [100]. Abuelezz and colleagues investigated the impact of PCSK9-inhibitor alirocumab in brain cholesterol metabolism, dyslipidemia, and neuroinflammation, which are markers of AD risk and pathology [99]. Results demonstrated a significant association between inhibition of PCSK9 and reduction of neuroinflammatory markers in rat models of AD [99]. Alirocumab-treated groups presented with increased LRP1 expression levels as well as reduced brain cholesterol levels, and neuroinflammatory marker expression including interleukin 1β (IL-1β), interleukin 6 (IL-6), and tumor necrosis factor α (TNF-α) [99]. Hippocampal BACE1, associated with Aβ clearance, was also reduced [99]. Taken together, these findings highlight the crosstalk between multiple metabolic pathways such as cholesterol homeostasis, neuroinflammation, AD pathology, and PCSK9 levels [99]. Zimetti et al. have also reported that PCSK9 levels were significantly higher in the cerebrospinal fluid of patients with AD compared to those without the disease [101]. Consistent with these clinical observations, *Pcsk9^−/−^*mice exhibited an upregulation of BACE1; the enzyme primarily responsible for producing amyloid plaques that is known to exacerbate AD [102]. As such, overexpression of PCSK9 reduced BACE1 expression in mice. In contrast, Apaijai et al. reported that inhibition of PCSK9 protected against dendritic spine loss by reducing amyloid plaque formation and neuroinflammation [103]. Similarly, Liu et al. also demonstrated that neuronal PCSK9 is not involved in the degradation of the LDLR or interaction with BACE1 in mice [104]. Results of the pivotal phase 3 clinical trials evaluating the efficacy of PCSK9 inhibition using monoclonal antibodies have concluded that there is no significant association between PCSK9 inhibition or lowered LDLc and neurocognitive decline. It is noteworthy, however, that a longer timepoint may be needed to observe the impact of this relatively new treatment on AD, which can take a lifetime to develop. Overall, as an important modulator of lipid homeostasis, it is possible that PCSK9 plays a role in neurodegenerative diseases, but at present this role remains unclear.

## 7. PCSK9 and Inflammatory Disease

In humans, dyslipidemia usually accompanies macrophage infiltration and the activation of numerous inflammatory pathways involved in the exacerbation of atherosclerosis and vascular plaque formation [105]. Recent studies have investigated the involvement of PCSK9 in several inflammatory processes in the context of atherosclerosis. Clinically, elevated white blood cell count is the primary marker of pro-inflammatory processes. In epidemiological studies, circulating levels of PCSK9 were positively associated with white blood cell, neutrophil, and lymphocyte count analyzed in both univariate and multivariate settings from patients with coronary artery disease (CAD) in China [106,107]. Similarly, fibrinogen—a pro-inflammatory marker known to be a blood coagulation glycoprotein—was positively associated with plasma PCSK9 levels from a cross-sectional study in patients with angiographically verified CAD [108].

Others have also shown that PCSK9 enhances the activation of several pro-inflammatory pathways. Tang et al. demonstrated that PCSK9 overexpression in oxLDL-treated macrophages promoted the production of oxLDL-induced cytokines, upregulated Toll-like receptor 4 (TLR4) expression, and increased nuclear factor-κB (NF-κB) nuclear localization [109]. Consistent with these observations, silencing of PCSK9 expression attenuated the production of pro-inflammatory cytokines by reducing both the nuclear localization of NF-κB and attenuating IκBα degradation. Others have used lipopolysaccharide (LPS), an agent well-known to trigger the NF-κB signaling pathway, to demonstrate its PCSK9-dependent pro-inflammatory effect [110]. Feingold et al. reported that LPS treatment in mice led to an increase in circulating PCSK9 levels, which was also accompanied by a reduction in hepatic LDLR expression and increased circulating LDL-cholesterol [111]. To strengthen this notion, Walley et al. demonstrated that human LOF PCSK9 variants were associated with improved survival in patients with septic shock, and that *Pcsk9*^−/−^ mice were protected from the pro-inflammatory cytokine response following treatment with LPS [86]. Given these observations, PCSK9 is clearly involved in the inflammatory response. The specific role of PCSK9 in this process, however, merits additional investigation.

## 8. Conclusions

PCSK9 was characterized nearly 20 years ago, and our understanding of its biology has evolved much during this period. Although the regulation of LDLc via the SREBP2-LDLR-PCSK9 axis remains its primary function, PCSK9 is clearly a multifaceted protein [11]. In recent years, our understanding of the alternative roles of PCSK9, as a protein that resides within secretory cells prior to secretion, has evolved. Both the secreted and intracellular forms of PCSK9 have been shown to play a role in a variety of diseases such as liver disease, CKD, neurodegenerative disease, (Figure 3).

In liver hepatocytes, PCSK9 has been shown to regulate a number of cell surface receptors, including those belonging to the LDLR family, as well as CD36. For this reason, as well as others, PCSK9 has now been shown to regulate hepatic lipid content and potentially affect the onset and progression of liver disease. Additionally, ER-resident PCSK9 has now been shown to regulate chaperone activity to influence the UPR and protect against ER stress. It is therefore pivotal to continue to develop our understanding of this important convertase family member to better harness the potential of the currently available PCSK9 inhibitors. Developing new modalities of PCSK9 inhibition, such as causing its retention in the ER, may also provide benefits that go beyond a reduction in CVD risk.

## Figures and Tables

**Figure 1 metabolites-12-00215-f001:**
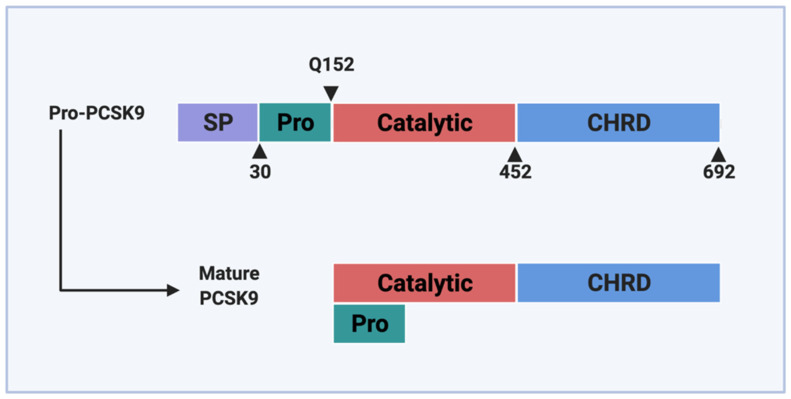
Domain structure of PCSK9. PCSK9 protein consists of a signal peptide (SP) that serves as a guide into the ER, a pro-domain (Pro) that undergoes self-cleavage during maturation, a catalytic domain, and a Cys-His-rich domain (CHRD) located at the C-terminus. Following the process of autocatalytic cleavage, the pro-domain remains associated with the catalytic domain and prevents further catalytic activity.

**Figure 2 metabolites-12-00215-f002:**
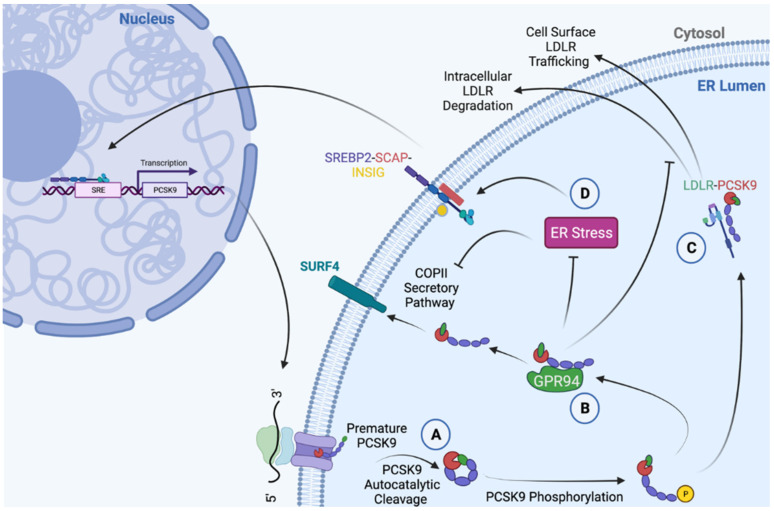
Summary of PCSK9 functions in the ER. (**A**) Mature PCSK9 is formed as a result of an autocatalytic cleavage event at position Q152↓, followed by the binding of the prodomain to the catalytic domain [2]. (**B**) The interaction between PCSK9 and ER chaperone, GRP94, mitigates PCSK9-mediated LDLR degradation [62], and increases the protein abundance of GRP94 [43,64]. (**C**) In addition to demonstrating co-chaperone functionality by increasing the abundance of GRP94 and protecting against ER stress, PCSK9 can act as a chaperone for pre-mature ER-resident LDLR and promote its trafficking to the cell surface [65]. (**D**) During conditions of ER stress, the abundance of ER-resident PCSK9 increases due to a reduction in secretion, as well as an increase in SREBP2-induced *de novo* synthesis [63]. Created with BioRender.com.

**Figure 3 metabolites-12-00215-f003:**
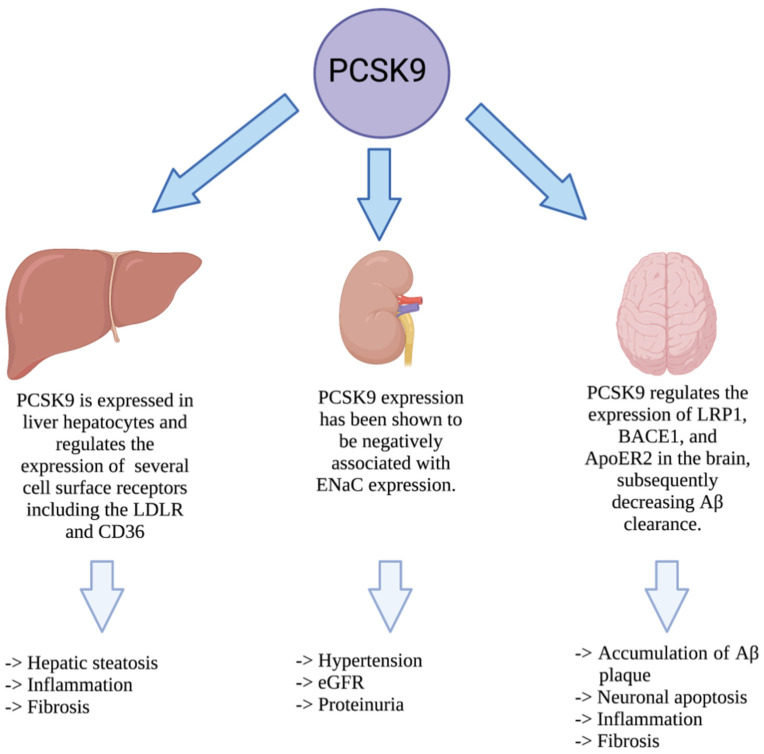
Impact of PCSK9 in non-cardiovascular diseases. PCSK9 is expressed in liver hepatocytes where it increases the stability of ER chaperones to protect against hepatic ER stress. Once secreted, PCSK9 can also contribute to hepatic steatosis, inflammation, and fibrosis. In the kidney, PCSK9 has been shown to affect hypertension, eGFR, and proteinuria. In the central nervous system, PCSK9 is also known to impact the accumulation of Aβ, leading to neuronal apoptosis, inflammation, and fibrosis.

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
