# Peer review of "The Emerging Roles of Intracellular PCSK9 and Their Implications in Endoplasmic Reticulum Stress and Metabolic Diseases"

_metabolites, 2022, doi:10.3390/metabo12030215_

Round 1
Reviewer 1 Report
The manuscript (review) of Lebeau et al. summarize the expression and cellular transport of wild-type PCSK9 and of several LOF’s and GOF’s of this protein. In addition the role of PCSK9 in several organs and corresponding diseases is highlighted. The manuscript provides a comprehensive overview of the current knowledge and is well written. The following points should be considered that may improve the understanding of PCSK9 for scientist not fully specialized in this field. 1. The authors are encouraged to provide a map of the protein sequence that display the different functional regions (epitopes) and the location of the various LOF’s and GOF’s mentioned in the manuscript. 2. It appears that many of the cited results are controversial and the reader is left to make up its own mind of the significance of these findings. The authors may add few sentences at the end of each chapter to state their own view and conclusions – if there are any. 3. There are currently several drugs on the market to reduce plasma PCSK9 quite effectively. This may have an important impact on the functionality of PCSK9 in different organs and in turn to potential unwanted side effects. A few sentences with that respect might be welcome.Author Response
The authors of this review article (metabolites 1580846) would like to thank the reviewers and editor for their efforts in increasing the breadth and quality of this work. Many revisions have been made in response to the comments, and we believe that the large majority have been addressed as requested. The responses to each of the comments are included below in bold type and line numbers refer to the “marked” version of the manuscript file.
- The authors are encouraged to provide a map of the protein sequence that display the different functional regions (epitopes) and the location of the various LOF’s and GOF’s mentioned in the manuscript.
The authors agree with the reviewer and have added a figure of the structure of PCSK9 (Figure 1 of the manuscript).
- It appears that many of the cited results are controversial and the reader is left to make up its own mind of the significance of these findings. The authors may add few sentences at the end of each chapter to state their own view and conclusions – if there are any.
The authors agree with the reviewer. Lines that now describe conflicting evidence are listed below.
- 80-81
- 250-254
- 298-303
- 334-338
- There are currently several drugs on the market to reduce plasma PCSK9 quite effectively. This may have an important impact on the functionality of PCSK9 in different organs and in turn to potential unwanted side effects. A few sentences with that respect might be welcome.
Additional context on the implications of currently approved anti-PCSK9 therapies in lines 250-254 as well as 298-303.

Reviewer 2 Report
The review manuscript submitted by Lebeau et al tilted ‘The emerging roles of intracellular PCSK9 and their implications in metabolic diseases’ investigates the importance of PCSK9 present inside the cell and its role in a variety of cellular pathways, metabolic diseases and LDLR-dependent and independent roles. Although the manuscript covers a wide range of topics, and is well researched, there are several flaws that need to be addressed. Therefore, we suggest a major revision.
Major comments:
- While the manuscript has a huge focus on the loss-of-function mutations of PCSK9 and the ER stress that it causes, the title or the abstract does not seem to suggest the same. This is highly misleading.
- Lines 57-58 mention the C-terminal domain. Please explain what the different modules of the C-terminal domain do.
- ‘the knockdown of sec24’ is a generalisation and is a misleading conclusion. The article cited shows that are four different types of sec24, out of which one has no effect. Please elaborate.
- After paragraph 2 in section 2, the topic shifts to mutations of PCSK9 and autocatalytic cleavage. Other factors that are involved in the secretion and transcription of PCSK9 are then described at the end of this section. If these paragraphs are moved to place after para 2, the reading would have a better flow.
- Lines 99-101 talks about GOF mutations that lead to reduced secretion of PCSK9. Is that true ?
- From lines 194, the writing shifts to LOF mutations and ER stress. Chapter 2 already discusses this in brief. Please combine these two, or make a separate section to talk about ER stress.
- Chapter 3 requires a concluding sentence to sum up the literature.
- Lines 240-242 should have been written earlier in the section.
- The last paragraph in chapter 4 talks about circulating as well as hepatic PCSK9. It would be nice to separate them as the paragraph is a bit confusing to read and understand.
- Chapter 5 also contains literature review about cardiovascular diseases. It would therefore be logical to combine CVDs and CKD literature and rewrite the section as cardio-renal diseases.
- BACE1 was discussed first, but the explanation about BACE1 comes later. Please either combine both, or write the explanation first.
- Conclusion chapter should draw a brief summary about the literature reviewed in the manuscript.
- Some abbreviations used are not explained in the text.
- Line 79 talks about druggable proteases and how PCSK9 is not one of them. This is not very clear.
- More Figures would be highly appreciated to visualize the role of PCSK9 in disease pathologies.
Minor comments:
- Some abbreviations used are not explained in the text.
- Rephrase or break the sentence that are in the lines 84-87.
- Rephrase or break the sentence that are in the lines 208-211.
- Rephrase the sentence that are in the lines 228-229.
- Rephrase the sentence that are in the lines 297-298.
- Change:
- Line 244: the extracellular -> extracellular
- Line 246: to lead to a reduction in -> reduce
- Line 253: an 11 -> 11
- Line 263: reinforcing -> reinforce
- Line 264: lipid metabolism homeostasis -> lipid metabolism/homeostasis
- Line 335: protected against -> protected them against
Author Response
- While the manuscript has a huge focus on the loss-of-function mutations of PCSK9 and the ER stress that it causes, the title or the abstract does not seem to suggest the same. This is highly misleading.
- The title has been reworded and highlights “endoplasmic reticulum stress” as a major topic of discussion. Because the majority of the LOF mutations in PCSK9 included in this review are intracellular forms of the protein, we consider “intracellular” to be inclusive of the LOF mutations.
- Lines 57-58 mention the C-terminal domain. Please explain what the different modules of the C-terminal domain do.
- Additional context included in lines 56-60
- ‘the knockdown of sec24’ is a generalisation and is a misleading conclusion. The article cited shows that are four different types of sec24, out of which one has no effect. Please elaborate.
- The authors agree with the reviewer and have add additional context to lines 71-73
- After paragraph 2 in section 2, the topic shifts to mutations of PCSK9 and autocatalytic cleavage. Other factors that are involved in the secretion and transcription of PCSK9 are then described at the end of this section. If these paragraphs are moved to place after para 2, the reading would have a better flow.
- The authors agree with the reviewer and have moved the text accordingly.
- Lines 99-101 talks about GOF mutations that lead to reduced secretion of PCSK9. Is that true ?
- We thank the reviewer for the attention to detail. We were in fact talking about LOF variants. This error has now been addressed (now line 131).
- From lines 194, the writing shifts to LOF mutations and ER stress. Chapter 2 already discusses this in brief. Please combine these two, or make a separate section to talk about ER stress.
- We agree with the reviewer that these sections could be combined, however, we believe that the section on PCSK9 as a co-chaperone should be a stand-alone. Section 2 discusses the mechanisms that lead to increased levels of intracellular PCSK9 (ie, LOF mutations), while section 3 focuses on the potential roles of intracellular PCSK9.
- Chapter 3 requires a concluding sentence to sum up the literature.
- The authors agree with the reviewer and have added additional context (lines 250-254)
- Lines 240-242 should have been written earlier in the section.
- The lines that discuss HFD have been moved forward in the paragraph (now lines 274-275).
- The last paragraph in chapter 4 talks about circulating as well as hepatic PCSK9. It would be nice to separate them as the paragraph is a bit confusing to read and understand.
- The authors agree with the reviewer that the paragraph discusses both intracellular and extracellular PCSK9. The authors prefer the structure of the text in its current form, however, with the first several paragraphs of chapter 4 on pre-clinical studies, and the last paragraph on clinical studies.
- Chapter 5 also contains literature review about cardiovascular diseases. It would therefore be logical to combine CVDs and CKD literature and rewrite the section as cardio-renal diseases.
- The authors agree with the reviewer and have changed the section heading.
- BACE1 was discussed first, but the explanation about BACE1 comes later. Please either combine both, or write the explanation first.
- Additional text was added to lines 396-397 for better describe the role of BACE in AD.
- Conclusion chapter should draw a brief summary about the literature reviewed in the manuscript.
- Additional detail has been added to the conclusion.
- Some abbreviations used are not explained in the text.
- Abbreviations were added for Sec24, HNF1a, Bag5, GALNT6, oxLDL, eGFR, ApoB, and AD.
- Line 79 talks about druggable proteases and how PCSK9 is not one of them. This is not very clear.
- The paragraph is not intended to make clear that PCSK9 is not druggable using small molecules. Rather, it is intended to present the challenges associated with the approach.
- More Figures would be highly appreciated to visualize the role of PCSK9 in disease pathologies.
- The authors agree with the reviewer and have now added two additional figures (now figs 1 and 3).
Minor comments:
- Some abbreviations used are not explained in the text.
- addressed
- Rephrase or break the sentence that are in the lines 84-87.
- The sentence has been rephrased
- Rephrase or break the sentence that are in the lines 208-211.
- The sentence has been rephrased
- Rephrase the sentence that are in the lines 228-229.
- The sentence has been rephrased
- Rephrase the sentence that are in the lines 297-298.
- The sentence has been rephrased
- Change:
- Line 244: the extracellular -> extracellular
- We respectfully disagree with the reviewer
- Line 246: to lead to a reduction in -> reduce
- addressed
- Line 253: an 11 -> 11
- addressed
- Line 263: reinforcing -> reinforce
- addressed
- Line 264: lipid metabolism homeostasis -> lipid metabolism/homeostasis
- addressed
- Line 335: protected against -> protected them against
- addressed

Round 2
Reviewer 2 Report
The authors have addressed my concerns appropriately and the manuscript is in my opinion acceptable for publication.